# Transcriptomic Profiling Revealed Signaling Pathways Associated with the Spawning of Female Zebrafish under Cold Stress

**DOI:** 10.3390/ijms23147494

**Published:** 2022-07-06

**Authors:** Guodong Ge, Yong Long, Guili Song, Qing Li, Zongbin Cui, Huawei Yan

**Affiliations:** 1Guangdong Provincial Key Laboratory of Microbial Culture Collection and Application, State Key Laboratory of Applied Microbiology Southern China, Institute of Microbiology, Guangdong Academy of Sciences, Guangzhou 510070, China; guodge@foxmail.com; 2State Key Laboratory of Freshwater Ecology and Biotechnology, Institute of Hydrobiology, Chinese Academy of Sciences, Wuhan 430072, China; longyong@ihb.ac.cn (Y.L.); guilisong@ihb.ac.cn (G.S.); qli@ihb.ac.cn (Q.L.); 3The Innovative Academy of Seed Design, Chinese Academy of Sciences, Beijing 100101, China

**Keywords:** reproduction, spawn, cold stress, RNA-seq, hormone

## Abstract

As one of the critical abiotic factors, temperature controls fish development and reproduction. However, the effects of low temperature on the transcriptional regulation of zebrafish reproduction remain largely unclear. In this study, the fecundity of zebrafish was examined after exposure to cold temperatures at 19.5 °C, 19 °C, 18.5 °C, or 18 °C. The temperature at 19 °C showed no significant influence on the fecundity of zebrafish, but temperature at 18.5 °C or 18 °C significantly blocked the spawning of females, suggesting the existence of a low temperature critical point for the spawning of zebrafish females. Based on these observations, the brains of anesthetized fish under cold stress at different cold temperatures were collected for high-throughput RNA-seq assays. Key genes, hub pathways and important biological processes responding to cold temperatures during the spawning of zebrafish were identified through bioinformatic analysis. The number of down-regulated and up-regulated genes during the temperature reduction from egg-spawning temperatures at 19.5 °C and 19 °C to non-spawning temperatures at 18.5 °C and 18 °C were 2588 and 2527 (fold change ≥ 1.5 and *p*-value ≤ 0.01), respectively. Venn analysis was performed to identify up- and down-regulated key genes. KEGG enrichment analysis indicated that the hub pathways overrepresented among down-regulated key genes included the GnRH signaling pathway, vascular smooth muscle contraction, C-type lectin receptor signaling pathway, phosphatidylinositol signaling system and insulin signaling pathway. GO enrichment analysis of down-regulated key genes revealed the most important biological processes inhibited under non-spawning temperatures at 18.5 °C and 18 °C were photoreceptor cell outer segment organization, circadian regulation of gene expression and photoreceptor cell maintenance. Furthermore, 99 hormone-related genes were found in the brain tissues of non-spawning and spawning groups, and GnRH signaling pathway and insulin signaling pathway were enriched from **down-regulated** genes related to hormones at 18.5 °C and 18 °C. Thus, these findings uncovered crucial hormone-related genes and signaling pathways controlling the spawning of female zebrafish under cold stress.

## 1. Introduction

Fishes are poikilothermic vertebrates whose body temperature varies depending on the ambient water temperature. Water temperature has a direct impact on growth, development and reproduction of fish [1]. The brain is the central organ of nervous system that controls key activities of the endocrine system [2]. Signals from the external environment, such as water temperature, can be sensed by the brain of fish to affect the reproductive processes through activating or inhibiting hormone synthesis and secretion, or alteration of hormone structure [3]. The hypothalamus–pituitary axis in the brain has the ability to process the signal of water temperature and further cope with changes in the whole body in ambient water temperature by stimulating target organs to secrete various hormones including thyroid hormone (TH), sex steroid, and glucocorticoid [4,5].

In fish, the reproductive process involves three basic steps: maturation, ovulation and spawning [6,7]. These steps are mainly regulated by the hypothalamic–pituitary–gonadal (HPG) axis consisting of three endocrine gland organs, the hypothalamus, the anterior pituitary gland (APG) and the gonads [8]. The HPG axis is mainly controlled by Kisepteptin (Kiss1) neurons that play pivotal roles in puberty onset through regulation of GnRH secretion in zebrafish [9], and GnRH neurons whose main function is to regulate the secretion of luteinizing hormone (LH) and follicular stimulating hormone (FSH) located in the hypothalamus [10]. When stimulated by external signals, LH and FSH are secreted to the extracellular environment and then transported through the bloodstream to the gonads of testicles or ovaries to regulate gamete formation and ovulation [11].

FSH regulates the spermatogonial proliferation and differentiation in Sertoli cells of males, while FSH plays a critical role in stimulating estrogen and inhibiting alpha-subunit (*inha*) production during folliculogenesis of females [12]. Expression of *inha* peaks during the full-grown stage of follicles and it acts as an endocrine hormone to trigger the final oocyte maturation and ovulation by stimulating LH production [13]. Follicle-stimulating hormone mainly stimulates the development of follicles by binding to follicle-stimulating hormone receptor, and luteinizing hormone regulates the maturation and ovulation of oocytes through luteinizing hormone receptor. A previous study has shown that the ovaries of *lhb*-deficient (lhb^−/−^) zebrafish are normal, but could not complete ovulation [14]. The anatomy of the homozygotes indicates that the ovaries contained a large number of oocytes that were about to mature, but did not develop into mature oocytes when they reached the stage of oocyte growth [14]. In addition, oocyte maturation and growth in zebrafish require the hormonal functions of 17α, 20β-dihydroxy-4-pregnen-3-one (17α, 20β-DP), a maturation-inducing hormone [15,16,17]. Moreover, the gonads also secrete hormones such as activin, statin, androgen and estrogen, which in turn act on the hypothalamus and pituitary to play a feedback regulating role [13].

In zebrafish, two forms of GnRH have been identified, namely GnRH2 and GnRH3, and they are expressed in the midbrain and olfactory bulb, respectively [18]. When GnRH3 neurons of adult zebrafish were ablated by laser pulses, oocyte development was arrested and average oocyte diameter was reduced [19]. However, reproduction is not compromised in *gnrh3*-knockout zebrafish (gnrh3^−/−^), and double genes (*gnrh2* and *gnrh3*) knockout zebrafish showed no major impact on reproduction [20]. The reproductive capability of spermatogenesis and folliculogenesis are not impaired in six mutant lines including *kiss1*^−/−^, *kiss2*^−/−^, *kiss1*^−/−^/*kiss2*^−/−^, *kissr1*^−/−^, *kissr2*^−/−^, and *kissr1*^−/−^/ *kissr2*^−/−^, indicating that kiss/kissr signaling is not absolutely required for zebrafish reproduction [21]. This shows us the complexity of regulatory mechanisms in zebrafish reproduction.

Besides the hypothalamic–pituitary–gonadal (HPG) axis, the reproductive system of zebrafish is tightly regulated by the hypothalamic–pituitary–thyroid (HPT) axis and hypothalamic–pituitary–adrenal (HPA) axis [22,23]. HPT is physiologically related to HPG and both of the axes work together in regulating reproductive functions [24]. The thyroid system mainly functions in the regulation of metabolism, growth and development of an individual. The secretion of thyroid hormones is known to affect the release of reproductive hormones such as LH, FSH and several steroid hormones, supporting the crosstalk concept between HPT and HPG [22]. The HPA axis is the complex set of interactions between the hypothalamus, pituitary gland, and adrenal gland [25], which is the major constituent of neuroendocrine system producing stress and mood responses and regulating the immune and reproductive systems. In zebrafish, the stress axis is known as the hypothalamus–pituitary–interrenal (HPI) axis [26].

In addition to hormones directly related to reproduction, other hormones such as growth hormone (GH), serotonin and melanin-concentrating hormone (MCH) also play important roles in the control of fish reproduction. Folliculogenesis has been arrested in females and spermatogenesis was delayed in males of *ghl*-mutant zebrafish [27]. The receptor of serotonin is mainly expressed in the preoptic area (5-HTr1aa and 5-HTr1ab), hypothalamus (5-HTr1aa, 5-HTr1ab and 5-HTr1bd) and ovary (5-HT2C) [28,29]. MCH can regulate the release of LH by activating pituitary or stimulating hypothalamic GNRH during reproduction [30]. Interestingly, GH synthesis in fish is modulated by water temperature and a high level of GH was found during the warmer seasons of the year [31].

Maturation of fish oocytes is the basis for spawning, but external conditions such as photoperiod [32], temperature and courtship [33] are necessary. However, it remains largely unclear how temperature controls the reproduction of fish. In this study, the effect of low temperature on zebrafish spawning was investigated. Key genes, hub pathways, and important biological processes that are closely associated with the spawning of female zebrafish were identified through transcriptomic analysis of zebrafish brain tissues under low temperatures. These findings are of great significance for further understanding the regulation of zebrafish spawning by the hormone-secreting axis in the brain.

## 2. Results

### 2.1. The Fecundity of Female Zebrafish under Low Temperatures

To examine the effects of low temperature on the fecundity of zebrafish, the spawning rates, numbers of eggs from each female and fertility rates of zebrafish at different low temperatures were measured. As shown by a representative assay in Figure 1A, females and males that were separately reared under normal conditions were put together at a ratio of one female to two males under five pre-set temperatures (28 °C, 19.5 °C, 19 °C, 18.5 °C and 18 °C) at 5:00 p.m. on the first day. Half an hour after light on at 8:30 a.m. in the next morning, seventeen females spawned at 28 °C (Figure 1B). After one and half hours from light on, only two zebrafish spawned at 19.5 °C and 19 °C, respectively (Figure 1B). After three and half hours, two zebrafish spawned at 18.5 °C and 18 °C, respectively. Four hours after light on, total numbers of zebrafish spawning at 28 °C, 19.5 °C, 19 °C, 18.5 °C, and 18 °C were 24, 13, 12, 3 and 2, respectively (Figure 1B). The average spawning rates at 28 °C, 19.5 °C and 19 °C were 82.76%, 65% and 70%, respectively, and showed no significant difference (Figure 1C). However, the average spawning rates of females at 18.5 °C and 18 °C significantly reduced to 15% and 10%, respectively (Figure 1C). During the experimental process, courtship behaviors of male zebrafish under cold temperature at 19.5 °C, 19 °C, 18.5 °C or 18 °C remained normal. The total number of spawned eggs from each female and the fertility rate of eggs from each female exhibited no significant changes at 28 °C, 19.5 °C, 19 °C, 18.5 °C and 18 °C (Appendix A).

Thus, low temperatures at 19.5 °C and 19 °C showed no significant effect on the spawning rates, numbers of eggs spawned by each female and fertility rates; however, a further decrease of 0.5 °C from 19 °C to 18.5 °C markedly inhibited the spawning rate of females, suggesting the existence of a low temperature critical point for the spawning of female zebrafish.

### 2.2. RNA-seq Analysis and Data Validation

To understand biological processes and signaling pathways controlling the formation of a low temperature critical point for the spawning of zebrafish females, twelve cDNA libraries of brains from spawning temperatures at 19.5 °C and 19 °C and non-spawning temperatures at 18.5 °C and 18 °C were constructed and subjected to high-throughput RNA-seq, followed by extensive bioinformatics analysis (Figure 2A). RNA-seq analysis generated 18.11–24.40 million (M) pairs of raw reads for each of the samples and about 72.32–77.77% of the processed reads were mapped to the reference genome of zebrafish (Figure 2B).

To validate the expression profiles from RNA-seq analysis, 17 genes were measured by qPCR. Primers used for qPCR sre listed in Appendix A. The gene of like-Sm protein 12 homolog b (*lsm12b*) was used as the reference gene [34]. As shown in Figure 2C, the Spearman bivariate correlation analysis revealed that data of RNA-seq and qPCR were significantly correlated (*p* < 0.00001, correlation coefficient = 0.8114); the fold changes of RNA-seq and qPCR for each gene are shown in Figure 2D, indicating the reliability of RNA-seq data.

### 2.3. Identification of Key Genes Associated with the Spawning of Females

The comparison of RNA-seq data resulted in four groups of differentially expressed genes (fold change ≥ 1.5 and a *p*-value ≤ 0.01) (Figure 3A) and the details of these genes are listed in Appendix A. There are 2588 up-regulated genes and 2527 down-regulated genes in group I, and 109 up-regulated genes and 83 down-regulated genes in group II. The differentially expressed genes in groups I and II logically contain critical factors controlling the spawning of females under low temperature.

Group III, resulting from the comparison of two non-spawning temperatures at 18.5 °C and 18 °C, contains 141 up-regulated and 125 down-regulated genes, and group IV, resulting from the comparison of two spawning temperatures at 19.5 °C and 19 °C, includes 213 up-regulated and 1295 down-regulated genes (Figure 3A). Thus, the differentially expressed genes in groups III and IV mainly resulted from the difference in water temperature and may not play key roles in female spawning.

We further performed the analysis of differentially expressed genes with Venn diagrams to identify key genes that are potentially required for the spawning of female zebrafish at low temperatures. As shown in Figure 3B,C, transcriptional changes of some differentially expressed genes in groups I and II (a, b, c, a’, b’, and c’) are likely associated with the spawning of female zebrafish under low temperature. The details and fold changes of genes obtained by Venn diagrams are displayed in Appendix A. Data of qPCR for three up-regulated genes (*actb1*, *atp2b3a* and *ralgapa2*) in groups I and II (a, b, and c) and three down-regulated genes (*mapre1b*, *pfkfb4b*, and *lgalsla*) in groups I and II (a’, b’, and c’) were highly consistent with those of RNA-seq (Figure 3D,E).

### 2.4. Enrichment of KEGG Pathways and Hub Genes Associated with Female Spawning

Up-regulated genes in groups I and II (a, b, and c) that are candidate key genes for the control of female spawning were mostly enriched in signaling pathways such as ribosome, RNA transport, and oxidative phosphorylation (Figure 4A and Appendix A). Down-regulated key genes in groups I and II (a’, b’ and c’) were mainly overrepresented in signaling pathways such as endocytosis, phosphatidylinositol signaling system and adrenergic signaling in cardiomyocytes (Figure 4B and Appendix A). In particular, the gonadotropin-releasing hormone (GnRH) signaling pathway that is considered to regulate the production through master hormones was significantly enriched in down-regulated key genes in groups I and II (a’, b’ and c’), indicating that the GnRH pathway was blocked in non-spawning groups at 18.5 °C and 18 °C (Figure 4B and Appendix A).

Since one gene is usually mapped to different signaling pathways, the Jaccard coefficient was introduced to calculate the distance between two signaling pathways according to the proportion of genes they shared, and the networks of up- and down-regulated KEGG pathways were obtained (Appendix A). Then, CytoHubba was used to identify the hub pathways in the networks. Among the signaling pathways enriched from up-regulated key genes in groups I and II (a, b, and c), the top three hub signaling pathways were metabolic pathways, carbon metabolism and pyruvate metabolism (Figure 4C and Table 1). Among the signaling pathways enriched from down-regulated key genes in groups I and II (a’, b’ and c’), the top three hub signaling pathways were GnRH signaling pathway, vascular smooth muscle contraction and C-type lectin receptor signaling pathway (Figure 4D and Table 2).

We also examined the hub genes for KEGG pathways with CytoHubba. The hub genes for up-regulated key genes were clustered into lactate dehydrogenase (*ldhba*, *ldhbb*), pyruvate kinase (*pkma*, *pkmb*, *pklr*), malate dehydrogenase (*mdh1aa*, *mdh1ab*), aldehyde dehydrogenase 9 family (*aldh9a1a.1*), pyruvate dehydrogenase (*pdhb*) and aldolase C, fructose-bisphosphate, a (*aldoca*) (Figure 5A and Appendix A). Among the hub genes, *pklr*, *pkma*, and *mdh1aa* in group I (a, b, and c) were up-regulated 3.2-fold, 2.4-fold, and 2.4-fold between the non-spawning and egg-spawning groups, respectively (Figure 5B). The hub genes for down-regulated key genes were clustered into calmodulin (*calm1b*, *calm2a*, *calm3a*, *calm3b*), protein kinase C (*prkcab*, *prkcba*, *prkcdb*), Ras homolog gene family (*rhoaa*), guanine nucleotide binding protein (*gnaq*) and v-src avian sarcoma viral oncogene homolog (*src*) (Figure 5C, and Appendix A). Among those hub genes, *prkcab*, *gnaq*, and *prkcdb* in group I (a’, b’ and c’) were up-regulated 3.7-fold, 3.3-fold, and 2.8-fold between the non-spawning and egg-spawning group, respectively (Figure 5D).

These data indicate that the metabolism signaling pathways were activated and the gonadotropin-releasing hormone (GnRH) signaling pathways were blocked in the group of non-spawning temperatures at 18.5 °C and 18 °C.

### 2.5. GO Enrichment of Key Genes Associated with Female Spawning

Differentially expressed genes that are potentially associated with female spawning were analyzed with GO enrichment. A total of 210 GO terms for up-regulated key genes and 106 GO terms for **down-regulated** key genes were enriched based on the *p*-value ≤ 0.05 and count ≥ 5 (Appendix A). Representatives of GO term through REVIGO tool were displayed in Appendix A. GO terms were clustered into three hierarchies, including biological process (BP), molecular function (MF) and cellular component (CC).

Up-regulated key genes in groups I and II (a, b, and c) under non-spawning temperatures at 18.5 °C and 18 °C were overrepresented in GO biological processes including glycolytic process (GO:0006096), translation (GO:0006412), axon extension (GO:0048675) and response to oxidative stress (GO:0006979) (Figure 6A). The most overrepresented GO molecular function for key genes in groups I and II (a, b, and c) were endopeptidase activity (GO:0004175), translation initiation factor activity (GO:0003743), and RNA helicase activity (GO:0003724) (Figure 6A). For cellular components, the up-regulated key genes in groups I and II (a, b, and c) were mostly enriched into cytosolic large ribosomal subunit (GO:0022625), nuclear pore (GO:0005643), nuclear speck (GO:0016607) and endoplasmic reticulum exit site (GO:0070971) (Figure 6A).

Down-regulated key genes in groups I and II (a’, b’ and c’) under non-spawning temperature at 18.5 °C and 18 °C were overrepresented in GO biological processes including rRNA processing (GO:0006364), vesicle docking involved in exocytosis (GO:0006904) and protein transport (GO:0015031) (Figure 6B). The most overrepresented GO molecular functions for key genes in groups I and II (a’, b’ and c’) were GTP binding (GO:0005525), GTPase activity (GO:0003924) and syntaxin binding (GO:0019905) (Figure 6B). For cellular components, the down-regulated key genes were enrichment to endosome (GO:0005768), nucleolus (GO:0005730) and retromer complex (GO:0030904) (Figure 6B). Among those biological processes, the most important processes inhibited when the temperature dropped from egg-spawning temperatures at 19.5 °C and 19 °C to non-spawning temperatures at 18.5 °C and 18 °C were photoreceptor cell outer segment organization (GO:0035845), G1/S transition of mitotic cell cycle (GO:0000082), circadian regulation of gene expression (GO:0032922) and photoreceptor cell maintenance (GO:0045494), indicating their importance in regulating the spawning of female zebrafish (Table 3).

### 2.6. The Expression of Hormone-Related Genes Decreased under Low Temperature Exposure

A total of 99 genes related to hormones were detected in spawning females at 19.5 °C and 19 °C and non-spawning females at 18.5 °C and 18 °C by RNA−seq (Figure 7A and Appendix A). These genes were divided into four clusters according to their expression levels within different temperature groups. The expression levels of genes in cluster 1 (*n* = 23, 23.23%) fluctuated slightly in spawning females at 19.5 °C and 19 °C, and were higher than those in non-spawning females at 18.5 °C and 18 °C. The expression of genes in cluster 2 (*n* = 40, 40.40%) decreased at temperatures from 19.5 °C to 18.5 °C, and showed no further change at temperatures from 18.5 °C to 18 °C. The expression levels of genes in cluster 3 (*n* = 16, 16.16%) fluctuated and showed a rising pattern from 19 °C to 18.5 °C. The expression levels of genes in cluster 4 (*n* = 20, 20.20%) fluctuated greatly in females of four temperature groups and exhibited no alteration patterns, so genes in cluster 4 were not used for subsequent enrichment analysis.

Both the neuroactive ligand–receptor interaction and the calcium signaling pathway were enriched in clusters 1 and 2. However, GnRH signaling pathway and insulin signaling pathway were enriched in cluster 1 and cluster 2, respectively (Figure 7B and Appendix A). In cluster 1, four gens (*lhb*, *gnrhr1*, *gnrhr4* and *cga*) enriched in the neuroactive ligand–receptor interaction were mapped to the GnRH signaling pathway and two genes (*oxtr* and *lhcgr*) were mapped to the calcium signaling pathway (Figure 7C). In cluster 2, three genes (*trhr2*, *trhra* and *trhrb*) enriched in the neuroactive ligand–receptor interaction were simultaneously mapped to the calcium signaling pathway. The insulin signaling pathway and neuroactive ligand–receptor interaction were linked through the gene of *spx* (spexin hormone). These data indicate that neuroactive ligand–receptor interaction, GnRH signaling pathway and calcium signaling pathway are cross-linked pathways to play crucial roles in the reproductive process of female zebrafish under low temperature through the alteration of hormone-related genes. Moreover, seven down-regulated key genes (*gh1*, *lhb*, *pmchl*, *pmch*, *cga*, *crhr2*, *ghrhrb*) were identified by Venn diagram. These genes were shared in the group of down-regulated genes (a’, b’ and c’) and cluster 1 and 2 (Figure 7D), suggesting their potential involvement in the spawning behaviors under different temperatures. Heat maps indicate that the expression levels of these key genes are lower under cold temperatures at 18 °C and 18.5 °C than those under cold temperatures at 19 °C and 19.5 °C (Figure 7E). RT-qPCR data of two representative **down-regulated** key genes, *ghl* (Figure 7F) and *lhb* (Figure 7G) are highly consistent with RNA-seq data

## 3. Discussion

Water temperature is a key signal for ovulation and spawning of fish, and broodfish are extremely sensitive to water temperature during the spawning period [35]. The spawning of some freshwater carps requires a water temperature above 18 °C [36]. However, the molecular mechanisms underlying the control of critical low temperatures for spawning of female fish remain largely unclear. In this study, we found that water temperature at 19 °C is the critical low temperature for the spawning of female zebrafish, since a decrease of 0.5 °C in water temperature from 19 °C to 18.5 °C markedly blocked the spawning of females. We further performed the transcriptional profiling of female brains with RNA-seq to identify key genes, hub pathways and important biological processes that are likely responsible for the spawning at 19.5 °C and 19 °C, and non-spawning at 18.5 °C and 18 °C.

The most important hub pathways of KEGG enrichment results were GnRH and insulin signaling pathways, whose activities are inhibited under cold stress at the non-spawning temperatures. GnRH secretion from the hypothalamus acts upon its receptor in the anterior pituitary to regulate the production and release of the gonadotropins, LH and FSH [13]. GnRHR, the receptor of GnRH, is coupled to Gq/11 proteins to stimulate the release of intracellular calcium for activating the intracellular calcium signaling pathway [37]. These signaling pathways are essential during folliculogenesis [12]. In addition, GnRH and insulin pathways were also significantly enriched in the gene set of cluster1 that decreased significantly at non-spawning temperatures. A previous study showed that light and pheromone-sensing neurons (ASJ) regulate cold acclimation through insulin signaling pathway in *Caenorhabditis elegans* [38]. Oocytes are naturally arrested at G2 of meiosis I, and exposure to either insulin/IGF-1 or the steroid hormone progesterone breaks this arrest and induces resumption of the two meiotic division cycles and maturation of the oocyte into mature fertilizable eggs. This process is also termed oocyte maturation [39]. Therefore, activities of GnRH and insulin signaling pathways are likely involved in the control of female spawning under cold stress.

A total of 99 hormone-related genes were differentially expressed between spawning and non-spawning females, which are mainly mapped to neuroactive ligand–receptor interaction, GnRH signaling, calcium signaling and insulin signaling pathways. Among these hormone-related genes, the most down-regulated genes were *gh1* (growth hormone 1) and *pmchl* (pro-melanin-concentrating hormone, like). These two genes were also found in the gene set of cluster 1, whose transcriptional expression decreased significantly from egg-spawning temperatures at 19.5 °C and 19 °C to non-spawning temperatures at 18.5 °C and 18 °C.

In addition to stimulating growth, cell reproduction and cell regeneration in humans and other animals, GH can stimulate the production of IGF-1 and increase the concentration of glucose and free fatty acids [40]. GH is critical in tissue regeneration, including in nerve liver in rats [41], skeletal muscle in adult rats [42], bone in rat calvarial defects [43], liver in mice [44] and hair cells in zebrafish inner ear [45]. A recent study found that zebrafish GH mutant females showed a severe dysfunction of gonadal development with ovarian folliculogenesis being arrested for months and then entering vitellogenic growth; the sperm isolated from the mutant testes developed normally but the adult mutant males could not breed with wild-type females [27]. MCH plays a key role in energy homeostasis and has a modulatory role with the release of LH either by directly acting on the pituitary gland or indirectly by affecting GnRH in the hypothalamus [30]. Thus, the main reason for non-spawning at the temperature of 18.5 °C and 18 °C is due to the inhibited expression of various hormones, especially GH and MCH. These hormones are likely playing key roles in determining the critical low temperature of female zebrafish spawning.

Calmodulin-encoding genes (*calm1b*, *calm2a*, *calm3a*, *calm3b*) were also identified as the hub genes, which were both mapped to the GnRH and insulin signaling pathways. A previous study noted the importance of calmodulin-mediated signaling pathways in plant response to cold stress [46]. It remains unclear how these signaling pathways regulate the reproduction of zebrafish under cold stress. Another family of hub genes was protein kinase-encoding genes including *prkcab*, *prkcba* and *prkcdb*, which were **down-regulated** and both mapped to GnRH signaling pathway and vascular smooth muscle contraction. PKC is a family of protein kinases that play important roles in several signal transduction cascades through controlling the function of other proteins by phosphorylation [47]. Detailed illustrations of the PKC family on cold acclimation and reproduction of fish need future investigations.

GO analysis indicated that the most representative GO terms were the glycolytic process for up-regulated key genes and rRNA processing for down-regulated key genes. There were 17 GO terms related to glycolysis enriched under cold stress in the non-spawning group and the mRNA level of genes encoding proteins in the energy metabolism cascades were up-regulated. Glycolysis is a metabolic pathway that converts glucose into pyruvic acid to release free energy [48]. Previous studies have reported that cellular ATP and ADP concentrations increased with decreasing temperature [49,50]. These biological processes appear to be closely associated with acclimation of fish under cold stress.

Light plays an important role in controlling the spawning of zebrafish [32] and the circadian rhythm plays an important role in regulating the function of photoreceptors cells [51]. In this study, photoreceptor cell outer segment organization was the most representative GO term of down-regulated key genes and circadian regulation of gene expression and photoreceptor cell maintenance were also significantly enriched by down-regulated key genes. In addition, there were five GO terms related to protein transport enriched by down-regulated key genes. Protein transport involves the secretion of proteins such as hormones from the intracellular compartment to extracellular space. Therefore, these biological processes are likely involved in the control of non-spawning female zebrafish since their activities were blocked at the non-spawning temperature.

Some of the GO molecular functions were overrepresented by the up-regulated key genes, such as endopeptidase activity, translation initiation factor activity, RNA helicase activity and transcription coregulator activity. It is likely that more proteins were required to maintain normal molecular functions because the activities of biological processes associated with these proteins were inhibited under cold stress. Moreover, GO molecular functions of “binding” were enriched by the down-regulated key genes, including GTP binding, syntaxin binding, ubiquitin protein ligase binding, phosphatidylinositol binding and magnesium ion binding. Molecular functions of “binding” play important roles in transmitting signals from outside a cell to its interior [52]. Our findings indicate that signal transduction related to these molecular functions is limited under non-spawning temperatures.

## 4. Materials and Methods

### 4.1. Animals and Experimental Conditions

The AB strain of zebrafish (*Danio rerio*) was raised and maintained under standard laboratory conditions (a photoperiod of 12 h light:12 h dark cycle; temperature at 28 °C). Adult zebrafish were fed twice daily with live brine shrimp and frozen bloodworms. Before the cold stress experiment, the adult zebrafish were naturally mated to ensure the ability of egg-spawning. Female zebrafish possessing full-grown mature oocytes were selected for the cold stress experiment [53].

In each experiment, one female and two males were chosen randomly and paired at the pre-set temperatures of 28 °C, 19.5 °C, 19 °C, 18.5 °C or 18 °C. The experiment started at 5:00 p.m. and lights were turned off at 8:00 p.m. on the first day; the entire process is shown in Figure 1A. Transparent baffle (TB) for the separation of two males and one female was removed at 8:30 a.m. when the light was turned on the next day, followed by counting the number of egg-spawning zebrafish at each temperature every half hour until 12:30 p.m. Each experiment contained at least three independent biological replicates. Water temperature was precisely controlled by Immersion Circulators and Coolers (PC200 Immersion Circulators, Thermo Fisher Scientific).

### 4.2. Tissue Collection and RNA Extraction

At the end of the experiment, four zebrafish were anesthetized to isolate the whole brain tissues and used as one sample. There were four temperature groups (19.5 °C, 19 °C, 18.5 °C, 18 °C) and each temperature group had three independent biological replicates. Thus, a total of twelve samples including six samples of egg-spawning temperature groups at 19.5 °C and 19 °C and six samples of non-spawning temperature groups at 18.5 °C and 18 °C were collected for RNA extraction.

### 4.3. Library Construction and High-Throughput RNA-Sequencing

The experiment contained 12 samples from four temperature groups (19.5 °C, 19 °C, 18.5 °C, 18 °C) and each temperature group had three independent biological replicates. Thus, 12 sequencing libraries were constructed and sequenced. Library construction and high-throughput RNA-sequencing (RNA-seq) were performed by experts in the Analytical and Testing Center at the Institute of Hydrobiology, Chinese Academy of Sciences (http://www.ihb.ac.cn/fxcszx/ (accessed on 1 October 2021)). The methods for sample quality analysis, preparation of RNA library and RNA-seq were described previously [54].

### 4.4. Bioinformatics Analysis

The raw reads were preprocessed to trim and filter low-quality data (Q < 20) using PRINSEQ (version 0.20.4) [55]. The cleaned and polished data for paired reads were extracted using Pairfq (version 0.14.4) [56]. These high-quality clean reads were then mapped to the reference genome (*Danio rerio* GRCz11) obtained from the NCBI assembly database (http://www.ncbi.nlm.nih.gov/assembly/GCF_000002035.6 (accessed on 15 November 2021)) using HISAT (Hierarchical indexing for spliced alignment of transcripts) (version 2.1.0) [57] to get the BAM formation of the aligned files. Then, the counts of reads were summarized using the read summarization program featureCounts [58] after using the samtools set of the command line to convert the binary BAM files into SAM files. These counts were used for gene differential expression analysis using the Bioconductor edgeR package [59]. Low abundance genes (number of summed reads < 10) were filtered before differential expression analysis. Genes with a fold change ≥ 1.5 and a *p*-value ≤ 0.01 were considered to be differentially expressed. Differentially expressed genes in each group were classified with Venn diagrams by the online tool (https://bioinfogp.cnb.csic.es/tools/venny/, v2.1 (accessed on 20 November 2021)) to identify key genes.

KOBAS 3.0 was performed for the Kyoto Encyclopedia of Genes and Genomes (KEGG) and Gene Ontology (GO) enrichment analysis of key up- and **down-regulated** genes [60]. The dot plots for KEGG enrichment results and bar plots for GO enrichment results (*p*-value ≤ 0.05) were generated by ggplot2 in R-studio (http://www.rstudio.com/ (accessed on 1 December 2021)). The R package (proxy, version: 0.4-26) was used to calculate the Jaccard coefficients between two KEGG signaling pathways based on the number of shared genes from the enrichment analysis results, and the network diagrams were created by Cytoscape (version: 3.8.2) (Institute for Systems Biology, Seattle, WA, USA) software. Cytoscape plug-in cytoHubba [61] was used to analyze hub signaling pathways and genes by MCC (Maximal Clique Centrality) method and we exported the visualization. REVIGO tool (http://revigo.irb.hr/ (accessed on 20 December 2021)) [62] was used to cluster and prune GO terms based on *p*-value obtained from KOBAS 3.0.

### 4.5. Signaling Pathways of Hormone-Related Genes

Transformed expressions based on TPM (transcripts per million) of all genes related to hormones identified in four groups were clustered using the fuzzy c-means (FCM) clustering algorithm implemented in the Bioconductor Mfuzz package [63]. According to the results of clustering, Cytoscape (v3.7.1) [64] plugin STRING [65] was used to analyze the protein interaction network and KEGG enrichment of genes in cluster 1 and cluster 2. Rstudio was used to visualize the results of KEGG enrichment analysis with bar charts.

### 4.6. Quantitative Real-Time PCR

Quantitative real-time PCR (qPCR) was performed as previously described [66] to validate the results of RNA-seq. The primers were designed using Primer Premier 6.0 (PREMIER Biosoft International, San Francisco, USA) software. Since the subunit S11 of the ribosomal gene *lsm12b* was not found to be differentially expressed in RNA-seq data among samples, it was selected as the reference gene for the normalization of gene expression as described previously [34]. qPCR data analysis was performed following the protocol of Hellemans et al. [67]. The sequence IDs, gene names and the length of amplicons are listed in Appendix A.

### 4.7. Statistical Analysis

Statistical analysis was performed with Microsoft Excel (Microsoft Office 2013, Microsoft, Redmond, WA, USA) software for windows. The data for spawning rates of zebrafish were statistically analyzed with the independent-samples *t*-test. The correlation of data between RNA-seq and qPCR was analyzed using the Spearman’s rho test.

## 5. Conclusions

In this study, we found the existence of a low temperature critical point for the spawning of zebrafish females. High-throughput RNA-seq assays of the brains of anesthetized fish under cold stress uncovered key genes, hub pathways and biological processes for the control of female zebrafish spawning under cold stress, such as genes encoding GH and MCH, signaling pathways of GnRH, insulin and PKC, biological processes of photoreceptor cell maintenance, circadian, glycolytic process, energy metabolism cascades, protein transport and molecular functions. These findings provide crucial clues for further dissecting the roles of key hormones and signaling pathways in the regulation of fish spawning.

## Figures and Tables

**Figure 1 ijms-23-07494-f001:**
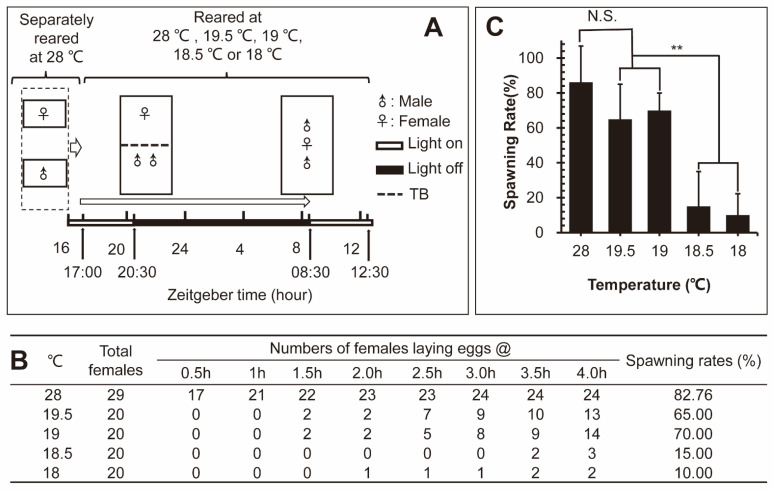
The fecundity of the zebrafish at different temperatures. (**A**) The schematic diagram for the fecundity of the zebrafish experimental process. (TB: transparent baffle). (**B**) The number of spawning zebrafish increased over time and spawning rates of female zebrafish at different temperatures. (**C**) Comparison of female spawning rates at different temperatures. (** *p* < 0.01; N.S.: no significance).

**Figure 2 ijms-23-07494-f002:**
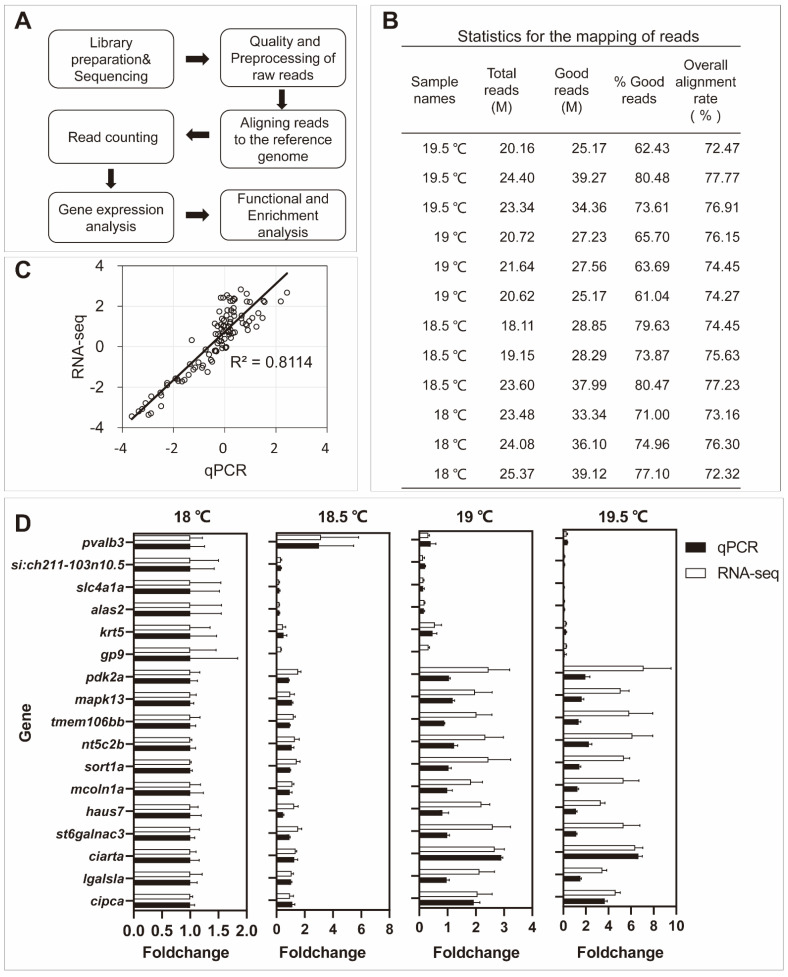
The workflow of RNA−seq data analysis. (**A**) Main steps of RNA−seq data analysis in the study. (**B**) Statistics for mapping of reads in four groups. The sample names used the experimental temperatures. Temperatures at 19.5 °C and 19 °C are egg-spawning groups and temperatures at 18.5 °C and 18 °C are non-spawning groups. (**C**) Validation of RNA−seq data with quantitative RT−PCR (qPCR). Log2 fold changes of RNA−seq data for gene expression were plotted against those of qPCR data. The reference straight line in black indicates the linear relationship between the results of RNA−seq and qPCR. (*p* < 0.00001, correlation coefficient = 0.8114). (**D**) Fold changes and standard error of mean (SEM) for 17 selected genes of RNA−seq were validated with qPCR.

**Figure 3 ijms-23-07494-f003:**
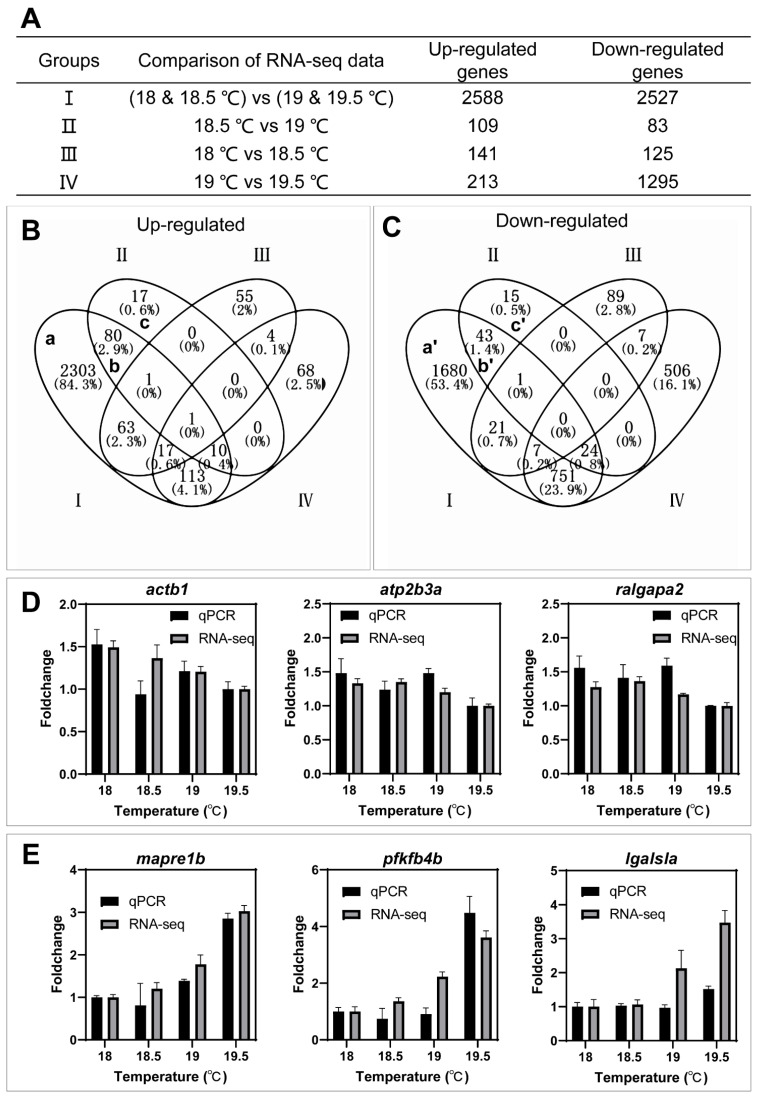
Identification of key genes associated with the spawning of females using Venn diagram. (**A**) The number of differentially expressed genes between groups of different temperatures (fold change ≥ 1.5 and *p*-value ≤ 0.05). (**B**) Up-regulated genes in different Venn groups to identify key genes between non-spawning and egg-spawning groups I and II (a, b, and c). (**C**) **Down-regulated** genes in different Venn groups to identify key genes between non-spawning and egg-spawning groups I and II (a’, b’ and c’). (**D**) Validation of RNA-seq data by qPCR for three genes from the group of up-regulated genes in groups I and II (a, b, and c). (**E**) Validation of RNA-seq data by qPCR for three genes from the group of **down-regulated** genes in groups I and II (a’, b’ and c’).

**Figure 4 ijms-23-07494-f004:**
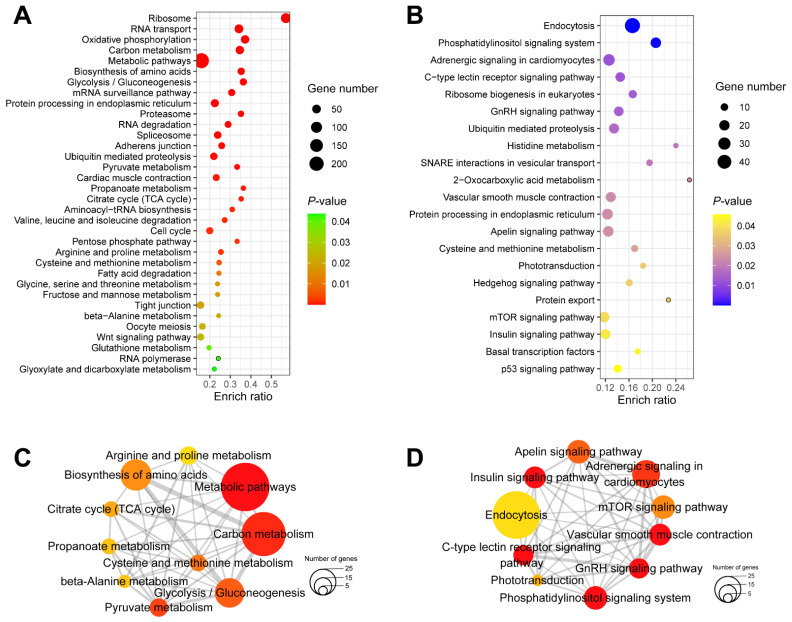
Hub signaling pathways from KEGG enrichment analysis of up- and **down-regulated** key genes. (**A**) Dot plot of KEGG signaling pathways for up-regulated key genes. (**B**) Dot plot of KEGG signaling pathways of **down-regulated** key genes. (**C**) Network of top 10 hub pathways for up-regulated key genes. (**D**) Network of top 10 hub pathways for **down-regulated** key genes. Nodes represent pathways and edges represent Jaccard similarity coefficients. Node color and size stand for the enrichment *p*-value and the number of genes in the pathway, respectively.

**Figure 5 ijms-23-07494-f005:**
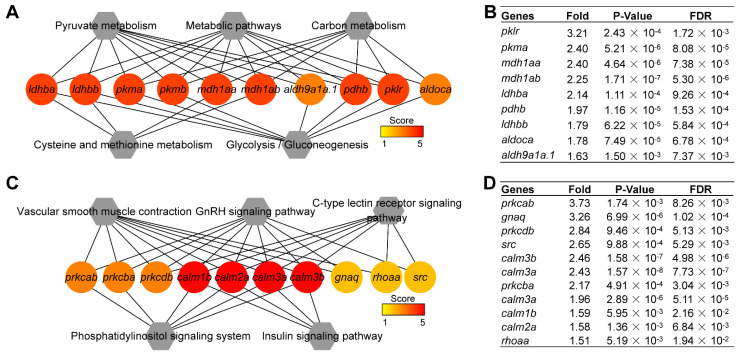
Hub genes within the top five hub pathways. (**A**) Network of 10 hub genes mapped to top five hub pathways enriched from up-regulated key genes. (**B**) The fold change of up-regulated hub genes in group I. FDR: false discovery rate (**C**) Network of 10 hub genes mapped to top five hub pathways enriched from **down-regulated** key genes. (**D**) The fold change of **down-regulated** hub genes in group I. Round stands for genes; hexagon represents pathways; edge means the gene mapped to the pathway in (**A**,**C**).

**Figure 6 ijms-23-07494-f006:**
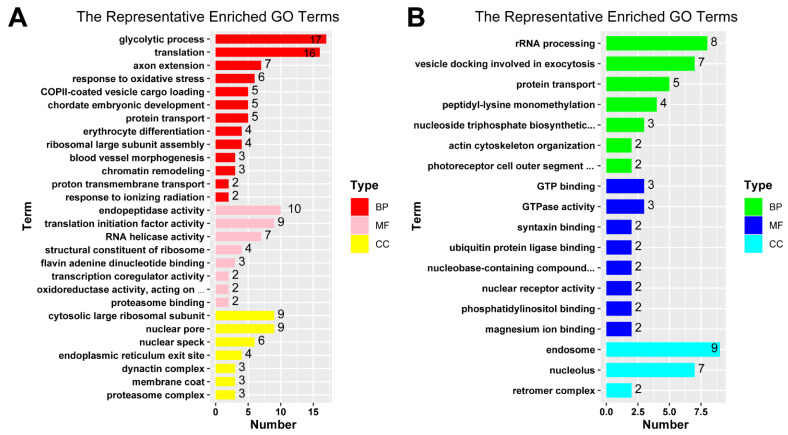
Bar plots for representative terms of GO enrichment analysis. GO terms redundancy were reduced by REVIGO tool to give a representative subset of terms for up-regulated key genes (**A**) and down-regulated key genes (**B**). Representative GO terms of different aspects of biology are shown in different colors, BP: biological process; MF: molecular function; CC: cellular component.

**Figure 7 ijms-23-07494-f007:**
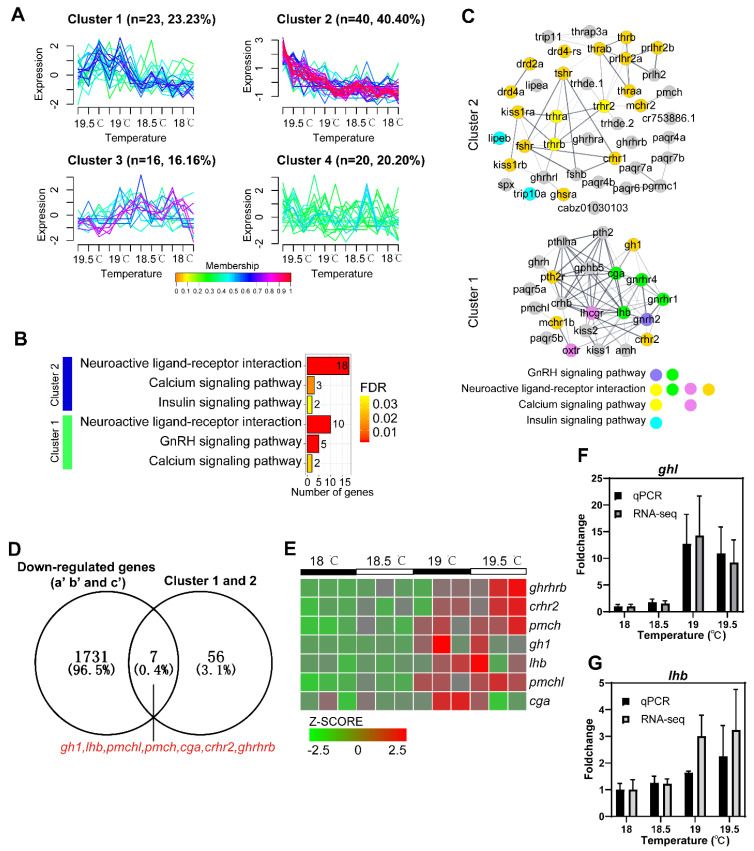
Details of hormone−related genes detected in the female zebrafish brains by RNA−seq. (**A**) Clustering analysis of hormone−related genes. (**B**) Bar plot of pathway enrichment analysis of genes in cluster 1 and cluster 2. False discovery rate (FDR) < 0.05. (**C**) Protein−protein interaction network of genes in cluster 1 and cluster 2. Nodes represent genes; edges stand for relationship of different genes; different color means different signaling pathways as shown at the bottom. (**D**) Venn diagram was used to identify down-regulated key genes that were shared in the group of down-regulated genes (a’, b’ and c’) and cluster 1 and 2. (**E**) Heat map of seven down-regulated key genes, presented as the normalization of TPM (z−score). (**F**,**G**) Fold changes and SEM of RNA−seq and qPCR for two representative down-regulated key genes *gh1* and *lhb.*

**Table 1 ijms-23-07494-t001:** Top 10 hub pathways of up-regulated genes ranked by MCC method.

Rank	Signaling Pathway	Score
1	Metabolic pathways	7691
2	Carbon metabolism	7686
3	Pyruvate metabolism	6600
4	Glycolysis/Gluconeogenesis	5760
5	Cysteine and methionine metabolism	5400
6	Biosynthesis of amino acids	3120
7	Citrate cycle (TCA cycle)	3006
8	beta-Alanine metabolism	3000
8	Propanoate metabolism	3000
10	Arginine and proline metabolism	2400

**Table 2 ijms-23-07494-t002:** Top 10 hub pathways of **down-regulated** genes ranked by the MCC method.

Rank	Signaling Pathway	Score
1	GnRH signaling pathway	10,800
1	Vascular smooth muscle contraction	10,800
1	C-type lectin receptor signaling pathway	10,800
1	Phosphatidylinositol signaling system	10,800
1	Insulin signaling pathway	10,800
6	Adrenergic signaling in cardiomyocytes	10,098
7	Apelin signaling pathway	10,092
8	mTOR signaling pathway	5766
9	Phototransduction	5040
10	Endocytosis	721

**Table 3 ijms-23-07494-t003:** The most important biological processes inhibited at non-spawning temperature.

Term ID	Description	Input Number	Background Number	*p*-Value
GO:0035845	photoreceptor cell outer segment organization	8	19	3.74 × 10^−4^
GO:0000082	G1/S transition of mitotic cell cycle	5	14	8.44 × 10^−3^
GO:0032922	circadian regulation of gene expression	8	37	1.19 × 10^−2^
GO:0045494	photoreceptor cell maintenance	5	22	3.69 × 10^−2^

## Data Availability

The data presented in this study are available in article and Appendix A. The sequencing data produced in this study have been deposited in the NCBI Sequence Read Archive (SRA) under the BioProject accession number PRJNA827490.

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
