# Peer review of "Transcriptomic Profiling Revealed Signaling Pathways Associated with the Spawning of Female Zebrafish under Cold Stress"

_ijms, 2022, doi:10.3390/ijms23147494_

Round 1

Reviewer 1 Report

The paper described their work on temperature and reproduction of zebrafish. The substance matter is interesting and is suitable to the journal. The findings are of interest to the readers of this journal. 

I have the following suggestions:

Major:

The authors may want to be careful not to be over interpreting. The main problem is in section 2.3 where the authors treated the very complex biological processes like a mechanical issue. Actually, the data indicate the largest numbers of DEGs were identified before the critical threshold of not being able to spawn. One cannot easily exclude those genes because they build up to the threshold. I would suggest the authors to tone down their comments, particularly in the following paragraph:

 "We further performed analysis of differentially expressed genes with Venn's diagrams to identify key genes that are specifically required for the spawning of female zebrafish under low temperature. As shown in Figures 3B and 3C, some of the differentially expressed genes in groups I and II (a, b, c, a', b', and c') are likely responsible for the spawning of female zebrafish under low temperature. The details and fold changes of genes obtained by Venn's diagrams were displayed in Table S4."

This is important because the authors need to function as critical scientists.

Minor, check English usage carefully, e.g.,

Line 41, eliminate the word “the” before growth.

Section 2.3., eliminate through Venn diagram. It sounds non-scientific.

Reviewer 2 Report

This is an interesting work studying the impacts of cold stress on zebrafish spawning behaviors. The authors determined the non-spawning temperatures and analyzed gene expression changes by RNA-seq. The manuscript contains large amounts of bioinformatic data, which are not very informative in the absence of experimental validation.

1. The RNA-seq data were not validated by qRT-PCR analyses. The authors should consider providing a main figure showing expression changes of key genes potentially involved in the spawning behaviors, under different tempratures.

2. The courtship behaviors of male zebrafish under cold stress are not clear. Changes in these behaviors may affect spawning of females. The authors need to address this important point.

3. Transcriptomic analyses were performed using female zebrafish. It will be interesting to examine and compare gene expression changes in male fish. At the least, some genes showing changes in females can be analyzed by qRT-PCR in males.

4. This study focused on gene expression in brain tissues. It may be worth to discuss other possible tissues or targets that may be affected by cold stress.

5. In the section of introduction, the authors need to present the rational to study cold stress on zebrafish spawning.

6. There is no functional analysis of genes possibly involved in spawning under cold stress. The study is descriptive and very correlative, thus the title is not appropriate. 

Round 2

Reviewer 2 Report

I am not completely satisfied with the revision because my main concern was not addressed.

One of my previous comments: “The authors should consider providing a main figure showing expression changes of key genes potentially involved in the spawning behaviors, under different temperatures”. 

It is not clear whether the 17 genes shown in figure 2C are “key genes potentially involved in the spawning behaviors”. Why did the authors select these genes? How are these genes related to spawning behaviors? There is no information on these genes in the main text. I would like to see a classical RT-qPCR result. This is not difficult because the authors can simply translate the data in table S1 into a graph.

Similarly, figures 3D and 3E compiling and analyzing more than 2500 genes from groups I and II are not informative. Again, how and why did the authors consider them as KEY genes, simply because they are up-regulated or down-regulated? Since there is no functional study in this work, the authors need to be cautious when interpreting their results. As above, it would be better to select a few genes and present a classical RT-qPCR graph.
